# MicroRNA as Possible Mediators of the Synergistic Effect of Celecoxib and Glucosamine Sulfate in Human Osteoarthritic Chondrocyte Exposed to IL-1β

**DOI:** 10.3390/ijms241914994

**Published:** 2023-10-08

**Authors:** Sara Cheleschi, Nicola Veronese, Serafino Carta, Giulia Collodel, Maria Bottaro, Elena Moretti, Roberta Corsaro, Marcella Barbarino, Antonella Fioravanti

**Affiliations:** 1Rheumatology Unit, Department of Medicine, Surgery and Neuroscience, Azienda Ospedaliera Universitaria Senese, Policlinico Le Scotte, 53100 Siena, Italy; saracheleschi@hotmail.com; 2Geriatric Unit, Department of Internal Medicine and Geriatrics, University of Palermo, Viale Scaduto, 90100 Palermo, Italy; 3Section of Orthopedics and Traumatology, Department of Medicine, Surgery and Neurosciences, University of Siena, Policlinico Le Scotte, 53100 Siena, Italy; cartaserafino@hotmail.com; 4Department of Molecular and Developmental Medicine, University of Siena, 53100 Siena, Italy; giulia.collodel@unisi.it (G.C.); elena.moretti@unisi.it (E.M.); r.corsaro@student.unisi.it (R.C.); 5Department of Medical Biotechnologies, University of Siena, 53100 Siena, Italy; mariaeusebia.bottaro@gmail.com (M.B.); marcella.barbarino@unisi.it (M.B.); 6Center for Biotechnology, Sbarro Institute for Cancer Research and Molecular Medicine, College of Science and Technology, Temple University, Philadelphia, PA 19122, USA; 7Independent Researcher, 53100 Siena, Italy; fioravanti7@virgilio.it

**Keywords:** microRNA, celecoxib, glucosamine sulfate, chondrocytes, osteoarthritis, NF-κB, inflammation, chondroprotection

## Abstract

This study investigated the role of a pattern of microRNA (miRNA) as possible mediators of celecoxib and prescription-grade glucosamine sulfate (GS) effects in human osteoarthritis (OA) chondrocytes. Chondrocytes were treated with celecoxib (1.85 µM) and GS (9 µM), alone or in combination, for 24 h, with or without interleukin (IL)-1β (10 ng/mL). Cell viability was determined using the 3-(4,5-dimethylthiazol-2-yl)-2,5-diphenyltetrazolium bromide (MTT) assay, apoptosis and reactive oxygen species (ROS) by cytometry, nitric oxide (NO) by Griess method. Gene levels of miRNA, antioxidant enzymes, nuclear factor erythroid (NRF)2, and B-cell lymphoma (BCL)2 expressions were analyzed by quantitative real time polymerase chain reaction (real time PCR). Protein expression of NRF2 and BCL2 was also detected at immunofluorescence and western blot. Celecoxib and GS, alone or in combination, significantly increased viability, reduced apoptosis, ROS and NO production and the gene expression of *miR-34a*, *-146a*, *-181a*, *-210*, in comparison to baseline and to IL-1β. The transfection with miRNA specific inhibitors significantly counteracted the IL-1β activity and potentiated the properties of celecoxib and GS on viability, apoptosis and oxidant system, through nuclear factor (NF)-κB regulation. The observed effects were enhanced when the drugs were tested in combination. Our data confirmed the synergistic anti-inflammatory and chondroprotective properties of celecoxib and GS, suggesting microRNA as possible mediators.

## 1. Introduction

Osteoarthritis (OA) is the most common musculoskeletal disease that causes significant disability among older adults and reduced the quality of life; findings from the recent Global Burden of Disease Study (GBD) reported an estimated prevalence in 2019 of 527.81 million which has dramatically increased by 113.25% from 1990 [1].

It is now understood, that OA is a joint disorder involving cartilage, synovial membrane, subchondral bone, ligaments, and capsule. Characteristic pathological findings include degeneration and loss of articular cartilage, synovitis, sclerosis of the subchondral bone, inflammation of the infrapatellar fat pad and instability of tendons and ligaments [2,3,4].

The breakdown of cartilage is the central hallmark of the disease; in physiological condition, the chondrocytes maintain an equilibrium between the biosynthesis and the degradation of matrix components. In OA, an altered imbalance of the chondrocytes activity leads to a progressive and gradual destruction of the tissue [5,6]. These alterations modify the biomechanical properties of the cartilage with further damage for the joint structure [7]. 

Moreover, pro-inflammatory factors, such as several cytokines [(IL)-1β, IL-6, IL-15, IL-17, IL-18, tumor necrosis factor (TNF)-α], prostaglandins, and reactive oxygen species (ROS), released by chondrocytes, synoviocytes, osteoblasts, or by the infrapatellar fat pad contribute to destroy cartilage structure and function [5,6,8]. Growing studies have underlined the key role of non-coding RNAs (ncRNAs) in regulating chondrocyte functions and in OA development and progression [9]. Among them, recently, microRNA (miRNA) have received extensive attention for their implication in OA pathogenesis [10,11,12,13,14]. Microarray analysis has identified different miRNA expression profile between normal and OA cartilage samples [15,16]; the evaluation of miRNA in synovial fluid, in serum, or in plasma has showed a significantly difference in patients with OA in comparison to healthy controls [17,18]. An aberrant expression of miRNA can influence chondrocytes phenotype and function, induce extracellular matrix degradation, apoptosis and oxidative stress, playing a pivotal role in OA pathogenesis [19,20,21,22]. Additionally, several miRNA affecting the expression of inflammatory mediators, such as the Interleukin-1β (IL-1β), which stimulates synovial inflammation, bone and cartilage destruction, characteristic features of OA disease [23,24].

To date, there is not a curative treatment for OA, thus the current strategies for its management have relied on a multimodal approach with a combination of pharmacological and/or non-pharmacological therapies focused on reducing pain and improving physical function [25]. Among the pharmacological options, high-quality prescription-grade glucosamine sulfate (GS), a symptomatic slow-acting drugs for osteoarthritis (SYSADOAs), has been recommended by the European Society for Clinical and Economic Aspects of Osteoporosis, Osteoarthritis and Musculoskeletal Diseases (ESCEO) working group for the treatment of OA due to its symptomatic and disease-modifying properties [26].

GS has shown to be effective in the regulation of chondrocytes and synoviocytes metabolism and in the reduction of IL-1β-induced negative activities [27,28,29,30,31].

Oral non-steroidal anti-inflammatory drugs (NSAIDs), selective and non-selective, are recommended to treat moderate to severe pain and inflammation in OA [26]. Celecoxib, a selective cyclooxygenase (COX)-2 inhibitor (COXIB), is one of the most frequently used symptomatic drug in clinical practice for its low toxicity, especially at the gastrointestinal and cardiovascular levels [23].

Accumulating evidence reported that celecoxib, besides its analgesic and anti-inflammatory properties, had also disease-modifying activities [32,33]. Indeed, different studies showed its regulation of cell metabolism, apoptosis, and oxidative stress in human OA chondrocytes, fibroblast-like synoviocytes, and subchondral bone osteoblasts [32,33,34,35,36,37,38,39]. Recently, it has been found the ability of celecoxib to restore the altered plasma expression of miR-155 and miR-146a in patients with knee OA [40], as well as the transcriptional levels of miR-29 and miR-34a in human gastric cancer cells and osteosarcoma lines [41,42].

In a previous report, we demonstrated, for the first time, the synergistic anti-inflammatory and chondroprotective effects of celecoxib and prescription-grade GS on human OA chondrocytes; this combination treatment exerted a protective role against the detrimental activities induced by IL-1β, mainly reducing inflammation and apoptosis, and regulating oxidant/antioxidant balance and cartilage turnover, via nuclear factor (NF)-κB pathway [43].

As an extension of these preliminary data, the present study aimed at investigating the potential regulatory effect of the celecoxib and GS, alone or in combination, on the expression profile of a pattern of miRNA in human OA chondrocytes exposed or not to IL-1β. In particular, this experience explored the possible ability of the two drugs to modulate the gene levels of miR-34a, miR-140, miR-146a, miR-155, miR-181a, miR-210, miR-375 and miR-let-7e, considering their involvement in OA inflammation, chondrocytes apoptosis and oxidative stress [9,13,19]. 

Moreover, we assessed the possible implication of the selected miRNA in mediating celecoxib and GS-induced effects on viability, apoptosis and oxidant system.

## 2. Results

### 2.1. Celecoxib and GS Modulate Apoptosis and Oxidant/Antioxidant System

In Figure 1 is reported the effect induced by celecoxib (1.85 µM) and GS (9 µM), alone or in combination, for 24 h, on the regulation of apoptosis ratio, oxidant/antioxidant system, nitric oxide (NO) release and viability, in OA chondrocytes stimulated or not with IL-1β.

The results showed that the treatment of the cells with celecoxib or GS alone didn’t cause any significant modification on apoptosis, redox balance and survival in comparison to control cultures (CTRL). The incubation of chondrocytes with celecoxib and GS in combination significantly reduced the apoptotic rate, up-regulated the gene expression of the anti-apoptotic marker B-cell lymphoma 2 (*BCL2*), while increasing cells viability, in comparison to CTRL (*p* < 0.05). The combined drugs significantly decreased ROS, NO production and the mRNA levels of nuclear factor erythroid 2 (*NRF2*) (Figure 1a–g). The latter represents the main transcription factor involved in redox homeostasis, and it was implicated in the pathogenesis of different pathological condition and in senescence events [44,45,46].

Conversely, IL-1β significantly induced the activation of apoptosis and oxidative stress, while reduced BCL2 levels and the percentage of survival (*p* < 0.01, *p* < 0.001); the negative trend of IL-1β was significantly counteracted by the pre-incubation of chondrocytes with celecoxib or GS (*p* < 0.05, *p* < 0.01), especially when used in combination (*p* < 0.05, *p* < 0.01, *p* < 0.001) (Figure 1a–g). Interestingly, the beneficial activities induced by the combined treatment in limiting IL-1β effects resulted more effective than each single compound (*p* < 0.05, *p* < 0.01) (Figure 1a–g).

To confirm the beneficial synergistic effect of the studied drugs on apoptosis and oxidant/antioxidant system, an immunofluorescence and a western blot analysis have been performed on the anti-apoptotic marker BCL2 and on the transcriptional factor NRF2 (Figure 2 and Figure 3). 

Figure 2a demonstrated that, at CTRL condition, the BCL2 signal was localized in the cytoplasm, and it appeared weak and diffuse in 48% of cells, while it appeared strong only in the 8% of chondrocytes; after the IL-1β stimulus, the strong localization was present in a percentage of 68% of cells and sometimes it appeared dotted (Figure 2b). The treatment of chondrocytes with celecoxib and GS in combination reduced the presence of cells strongly positive to BCL2 protein in comparison to the only IL-1β stimulus (30%) (Figure 2d).

The image reported in Figure 2e–h showed the different localization of NRF2 protein in the studied experimental conditions, less evident compared BCL2 protein. A diffuse weak or absent cytoplasmic distribution was detected at CTRL in most ofthe cells (79%) in which a clear signal was present in the 21% (Figure 2e). The incubation of chondrocytes with IL-1β determined an increase of number of cells with a strong localization (70%) (Figure 2f) that was reduced by the pre-treatment with celecoxib and GS in combination (35%) (Figure 2h).

No significant different localization of both antibodies was observed when the cells were incubated with the studied drugs, alone or in combination, compared to CTRL or to the stimulus of IL-1β (BCL2 Figure 2c: NFR2 Figure 2g). 

A similar trend was obtained on the densitometric quantification of the bands assessed at western blot analysis (Figure 3a–d). The results demonstrated that the stimulus of chondrocytes with IL-1β induced a significant increase of BCL2 and NRF2 protein expression (*p* < 0.05) in comparison to CTRL. Celecoxib and GS tested in combination significantly reduced the expression of BCL2 (*p* < 0.05) with respect to control condition, while no modifications on NRF2 were found. More interestingly, the combined effect of the studied drugs was able to counteract, in a significant manner, the negative activity of IL-1β on these proteins. 

### 2.2. MiRNA Profile Modulation Following Celecoxib and GS Treatment

Figure 4 showed the ability of celecoxib and GS to regulate the expression profile of a pattern of miRNA, involved in the pathogenesis of OA, modified by IL-1β.

The combination of the studied pharmacological compounds induced a significant reduction of *miR-34a*, *miR-146a*, *miR-181a*, and *miR-210* gene expression (*p* < 0.05), while no changes in *miR-140*, *miR-155*, *miR-375*, and *miR-let-7e* levels were found, with respect to CTRL (Figure 4a–g). As expected, IL-1β stimulus significantly up-regulated the transcriptional levels of *miR-34a*, *miR-146a*, *miR-155*, *miR-181a*, *miR-210*, and *miR-let-7e* (*p* < 0.01), while decreased those of *miR-140* and *miR-375* (*p* < 0.01). 

The pre-treatment of the cells with celecoxib or GS alone or in combination significantly limited the expression of *miR-34a*, *miR-146a*, *miR-181a*, and *miR-210* induced by IL-1β, in comparison to the incubation with the cytokine alone, with a particular enhancement when the drugs were tested simultaneously (*p* < 0.05, *p* < 0.01, *p* < 0.001) (Figure 4a,c,e,f). Interestingly, the combined use of the studied pharmacological compounds resulted to be more efficacious in counteracting IL-1β effects on these miRNA, than what was observed by each single treatment (*p* < 0.05) (Figure 4a,c,f).

Otherwise, *miR-140*, *miR-155*, *miR-375*, and *miR-let-7e* profiles appeared not to be influenced by the treatment with the drugs alone or in combination (Figure 4b,d,g,h).

To better understand the direct regulation of celecoxib and GS on miR-34a, miR-146a, miR-181a, and miR-210 expression profile, chondrocytes were transiently transfected with miRNA specific inhibitors for 24 h before the standard treatment with the drugs (Figure 5). 

Figure 5a corroborated the ability of the inhibitors to significantly reduce the gene expression of *miR-34a*, *miR-146a*, *miR-181a*, and *miR-210* (*p* < 0.01, *p* < 0.001) in comparison to CTRL cultures and negative control (NC). After miRNA silencing, the cells exposed to IL-1β showed a significant down-regulation of *miR-34a*, *miR-146a*, *miR-181a*, and *miR-210* compared to the stimulus with IL-1β alone (*p* < 0.05) (Figure 5b–e). Interestingly, the effects induced by IL-1β resulted strongly neutralized by the concomitant exposure of the cells to miR-34a, miR-146a, and miR-210 inhibitors and the drugs, in comparison to their relative NC (*p* < 0.05, *p* < 0.01) (Figure 5b,c,e). On the other hand, celecoxib and GS activities seem to be not dependent by the presence of miR-181a inhibitor (Figure 5d).

### 2.3. MiRNA Mediate the Effect Induced by Celecoxib and GS

To verify the potential role of miRNA as mediators of celecoxib and GS alone or in combination effects on viability, apoptosis and oxidant system, our cultures were simultaneously incubated with miR-34a, miR-146a, miR-181a, and miR-210 inhibitors and treated with the pharmacological compounds. 

Our results showed that the silencing of chondrocytes with miR-34a, miR-146a, miR-181a, and miR-210 inhibitors induced a significant reduction of apoptosis (*p* < 0.05, *p* < 0.01), an increase of viability (*p* < 0.05) while limited the production of ROS and NO (*p* < 0.05, *p* < 0.01), with respect to CTRL cultures and NC (Appendix A, Figure 6a–h, Figure 7a–h and Figure 8a–d). 

The stimulus with IL-1β caused opposite and detrimental effects increasing apoptosis and oxidative stress, which was significantly reduced by the pre-treatment of the cells with celecoxib and GS, or transiently transfected with miRNA specific inhibitors (*p* < 0.05, *p* < 0.01) (Figure 6a–h, Figure 7a–h and Figure 8a–d).

Interestingly, the co-incubation of the cells with miR-34a, miR-146a, miR-210 inhibitors and the pharmacological compounds significantly reinforced their inhibitory effect against IL-1β on the studied processes (*p* < 0.05, *p* < 0.01), than those induced by each single treatment (Figure 6a,b,d–f,h, Figure 7a,b,d–f,h and Figure 8a,b,d). Also in this case, celecoxib and GS did not alter miR-181a inhibitor activities on apoptosis and oxidative stress (Figure 6c,g, Figure 7c,g and Figure 8c).

### 2.4. NF-κB Involvement in Drugs-Induced Effects on miRNA

On the basis of the obtained results and the direct interaction between celecoxib and GS with NF-κB found in our previous study, we hypothesized miRNA as possible mediators of drugs’ effects, via NF-κB. At this regard, our cultures were pre-incubated with a specific NF-κB inhibitor (BAY 11-7082, IKKα/β) and, then, treated with the studied drugs and IL-1β (Figure 9).

An analysis at real time PCR was performed on miR-34a, miR-146a and miR-210, which resulted already modulated by celecoxib and GS, to explore their regulation in presence of NF-κB inhibitor. 

The data reported in Figure 9 showed a significant decrease in gene expression of *miR-34a*, *miR-146a* and *miR-210* after incubation of the cells with BAY 11-7082 with respect to CTRL and in particular to IL-1β stimulus (*p* < 0.05, *p* < 0.01) (Figure 9a–c). Very interestingly, miRNA down-regulation induced by BAY 11-7082 effect was significantly increased when the cells were concomitantly treated with celecoxib or GS (*p* < 0.05), in a more effective way beyond that caused by the treatment with the only drugs (Figure 9a–c). As expected, the simultaneous exposure of the cells to Cel and GS plus BAY 11-7082 resulted more efficacious in regulating miRNA expression than what was found by each single treatment (*p* < 0.05) (Figure 9a–c).

## 3. Discussion

This study aimed at deeper investigating the molecular mechanism underlying the beneficial properties of celecoxib and prescription-grade GS on human OA chondrocytes, based on the preliminary data obtained in our previous research [43]. In particular, we analyzed the regulatory effects of the two drugs, alone or in combination, on apoptosis, viability, and oxidant system, as well as on the expression profile of a pattern of miRNA involved in OA pathophysiology. In addition, we hypothesized miRNA as possible mediators of celecoxib and GS-induced effects.

The doses of celecoxib and GS used to conduct our experiments represent the most suitable concentrations to reproduce in vivo condition [47,48] and were selected in agreement with our prior data [43]; the treatment was applied for a period of 24 h based on the best results in terms of viability.

This study confirmed the favorable effects of celecoxib and GS on apoptosis, viability and redox balance, according to our preliminary report [43]. These processes are well known as important hallmarks associated to OA unset and progression [6,49]. Indeed, during the pathogenesis of the disease, inflammatory mediators, such as cytokines, can alter mitochondrial structure and function causing the failure in oxidant/antioxidant system, leading to an excessive ROS generation. This condition contributes to activate cartilage degradation and synovial inflammation, as well as the apoptosis signaling [6,50,51]; the latter is controlled and regulated through a balance between the members of the BCL2 family characterized by proteins with pro-apoptotic (i.e Bax, Bak) and anti-apoptotic roles, such as BCL2, which generally driving anti-apoptotic functions [51].

Our data showed an increased production of mitochondrial, extracellular ROS species and of NO, and an up-regulation of the transcriptional factor *NRF2*, in OA chondrocytes exposed to IL-1β; also, this negative stimulus raised the ratio of apoptotic cells, reduced the percentage of viability, with a concomitant decreased gene expression of the anti-apoptotic marker *BCL2*. These data are consistent with other evidence from the literature conducted on chondrocyte cultures [39,43,52,53]. 

Surprisingly, our immunofluorescence and western blot assays showed opposite findings on BCL2 protein expression, after IL-1β stimulus than those found at its mRNA levels. The apparent discrepancy among the results can be attributed to the complex interplay between the anti-apoptotic and pro-apoptotic members of BCL2 family during the apoptosis process. Indeed, it has been demonstrated that BCL2 exerts its role of anti-apoptotic protein by binding some pro-apoptotic molecules to inhibit the apoptosis signaling; once the binding among the proteins is dissociated, BCL2 can be phosphorylated in the cytoplasm and degraded by different kind of stimuli [51,54,55]. This can explain, in our opinion, the increased levels of BCL2 protein at the immunofluorescence analysis, since, for our experiments, we used an anti-BCL2 antibody not able to discriminate the active non-phosphorylated BCL2 protein from the inactivate phosphorylated form destined for degradation. 

Furthermore, we corroborated the ability of celecoxib and GS, tested alone or in combination, to limit the negative effect of IL-1β, decreasing ROS and NO release, *NRF2* expression, and cell death process, according to previous findings. In fact, different authors reported a reduction of ROS and NO production and antioxidant enzymes expression, induced by IL-1β, after treatment of human OA cartilage explants or chondrocytes with celecoxib and/or GS [39,43,56,57]. Similarly, both drugs limited the apoptosis rate and increased the anti-apoptotic marker expression, in IL-1β-stimulated OA cells [31,39,43,58].

The obtained results concerning the reduction of NRF2 levels after celecoxib and GS treatment seem to be in apparent discrepancy with some recent data from the literature which reports an increase of this transcriptional factor following treatment with pharmacological substances [45] or its reduced activity under oxidative stress condition [44,46]. However, it has been demonstrated that, in physiological conditions, NRF2 is maintained at low levels in the cytoplasm by its ubiquitination and degradation through proteasome machinery; stress signals, including pro-inflammatory cytokines, lead to NRF2 activation, through the release from its repressive cytosolic protein Kelch-like ECH associated protein1 (KEAP1), and its translocation into the nucleus, where it regulates the transcription of antioxidant response element-dependent genes [21,43,59,60]. This can explain the strong increase of NRF2 levels that we observed following IL-1β exposure, as well as its evident reduction in presence of the studied drugs.

These results confirm the anti-apoptotic and antioxidant activities of celecoxib and GS, especially when used in combination. However, their exact molecular mechanism and, thus, the beneficial properties of these drugs still remain not completely defined.

In the second part of the study, we verified the hypothesis that miRNA could represent possible mediator factors by which celecoxib and GS exert their effects on apoptotic process and oxidant system.

Accumulating evidence proved that some of the most important miRNA associated to OA pathogenesis are strong regulators of apoptosis and oxidative stress signaling [9,13,19]. In particular, miR-34a and miR-181a were found to increase cell death and reduce cell viability, as well as to modulate the production of ROS and NO, in different cell types [9,13,61,62,63,64]. Also, miR-146a has been recently recognized as an activator of these processes by its direct targeting on SMAD4 and NRF2 transcriptional factors, in human OA chondrocytes and synoviocytes [21,65,66]. In addition, miR-210 was found to be associated to OA due to its higher levels in synovial fluid of patients with knee OA compared to healthy individuals [67]. Furthermore, miR-210 contribute to chondrogenic differentiation, DNA damage response, and apoptosis in bone marrow mesenchymal stem cells and synovial fibroblasts [68,69]. 

In the present research, we showed the over-expression of *miR-34a*, *miR-146a*, *miR-181a*, and *miR-210* after the stimulus of OA chondrocytes with IL-1β, consistently with the results derived from other in vitro studies [70,71,72]. Interestingly, we also observed that the pre-incubation of our cells with celecoxib and GS significantly reduced the expression of the studied miRNA, counteracting the negative effect induced by IL-1β. This is the first report revealing the ability of these drugs to regulate miR-34a, miR-146a, miR-181a, and miR-210 profiles in human OA chondrocytes, and, especially, highlighting their potentiated effect when used simultaneously. Previous studies confirmed the relevance of celecoxib in controlling the gene expression of some miRNA involved in the progression of human malignancies has been documented. In particular Chen et al. [73], performing an analysis of miRNA expression profile in human colorectal cancer cells, reported that celecoxib limited the aberrant expression of 28 miRNA correlated to clinical stage of cancer, lymph node involvement and metastasis. More recently, other authors, through a miRNA microarray analysis on gastric cancer cells, proved the clinical relevance of celecoxib for the treatment of gastric cancer, fine-tuning the tumor suppressor miR-29 [41].

To attest the direct effect of celecoxib and GS on the studied miRNA, we carried out additional experiments of miRNA silencing. In fact, the transient transfection of our OA chondrocytes with miR-34a, miR-146a, miR-181a, and miR-210 inhibitors significantly reduced the apoptotic process, ROS generation and NO production, and increased the percentage of viability, especially preventing their activation induced by IL-1β. In a similar manner, previous research on OA chondrocyte and synoviocytes employing miRNA inhibitors attested a direct modulation of miR-34a, miR-146a and miR-181a on cell proliferation and apoptosis by targeting SIRT-1/p53 and on BCL2 signaling pathways [21,74]. The inhibition of these miRNA also limited the release of ROS species and the production of NO in different cell cultures, probably affecting NRF2 proteins and NF-kB pathway [21,63,74,75,76].

Intriguingly, we firstly demonstrated that the influence of miR-34a, miR-146a, and miR-210 silencing on apoptosis and redox balance was strongly reinforced in presence of celecoxib and GS. In support of our data Chen and co-authors [42] demonstrated the synergistic, additive effect of celecoxib and miR-34a in the regulation of cell viability, cell migration and invasion in osteosarcoma cell lines. 

As the last step of our research, we have tried to find out the regulatory network underlying the interaction between celecoxib and GS and miRNA, assuming that a key role could be played by NF-κB signaling pathway. A number of studies, even our previous one, demonstrated that these pharmacological compounds exert a protective role on cartilage metabolism and inflammation through the regulation of NF-κB proteins [28,32,43]. In fact, evidence from cancer cell lines and human chondrocytes demonstrated the ability of celecoxib in controlling apoptosis and oxidative stress processes through a direct effect on p50 and p65 subunits of NF-kB pathway [43,77,78]. GS exerts its role on NF-κB via an epigenetic mechanism, regulating the demethylation of specific CpG sites of DNA [79,80], responsible for the expression of redox- and apoptosis-related factors, in human articular chondrocytes [43,79,81]. In addition, several in vitro evidence highlighted a direct effect between some miRNA, including miR-34a, miR-146a, and miR-181a, and NF-κB signaling, which reflects a consequent modulation of the downstream genes controlled by the pathway [21,82,83]. 

According to the current literature, in our previous report we confirmed the direct combined effect of celecoxib and GS on NF-κB pathway in OA chondrocytes, demonstrating the capacity of the drugs to reduce the expression of p50 and p65 subunits [43]. Very interestingly, in the present paper, for the first time, we showed that the inhibition of NF-κB signaling in our cultures, induced a reduction of the expression profile of *miR-34a*, *miR-146a*, and *miR-210*. Intriguingly, this trend resulted more favorable and strengthen in the presence of celecoxib and GS in combination.

On the other hand, celecoxib and GS properties seem to be not influenced by miR-181a, indeed, no detectable changes on apoptosis or redox balance were observed in our results following miR-181a silencing. To the best of our knowledge, there is no evidence from the literature explaining this aspect, so it’s difficult to draw any conclusion in this sense; therefore, it’s reasonable to assume that miR-181a and celecoxib and GS act through two independent mechanisms on cell death and ROS modulation. 

## 4. Materials and Methods

### 4.1. Primary Cultures of Human OA Chondrocytes

Human OA articular cartilage was obtained from femoral heads of five patients with hip OA according to ACR criteria [84], subjected to total arthroplasty. The donors were two men and three women with age ranging from 61 to 76. The samples were provided by the Orthopaedic Section from the University of Siena (Italy), and OA grades was defined by Mankin score [85]. Authorization at using human specimens was permitted by the Ethic Committee of Azienda Ospedaliera Universitaria Senese (decision no. 13931/18), after receiving the signed informed consent from the donor. 

Chondrocytes were isolated immediately after surgery. Cartilage slices were aseptically dissected from the femoral heads and processed by a sequential enzymatic digestion using trypsin and type IV collagenase, as previously described [21]. Then cells were grown in Dulbecco’s Modified Eagle Medium (DMEM) (Euroclone, Milan, Italy) with phenol red and L-glutamine, supplemented with 10% fetal bovine serum (FBS) (Euroclone, Milan, Italy), and 200 U/mL penicillin and 200 µg/mL streptomycin (P/S) (Euroclone, Milan, Italy). After a test of viability with Tripan Blue (Sigma-Aldrich, Milan, Italy), primary cells at the first passage were employed for the experiments. 

### 4.2. Treatment Procedure

Human OA chondrocytes were plated in 6-well dishes at a starting density of 1 × 10^5^ cells/well and cultured in DMEM containing 10% FBS and 2% P/S, until reaching about 85% of confluence.

Prescription-grade crystalline glucosamine sulfate (Dona^®^) and celecoxib (Celebrex^®^) were provided by Meda Pharma SpA (Viatris group, Monza, Italy,). The powders were reconstituted in phosphate-buffered saline (PBS) (Euroclone, Milan, Italy) filtered and diluted in the culture medium (DMEM with 0.5% FBS and 2% P/S) to reach the final concentrations selected for the experiments. Celecoxib and GS were tested at the concentration of 1.85 µM and 9 µM, respectively, for 24 h, in agreement with previous studies [43,47,48]. The treatment was conducted in presence or not of IL-1β (10 ng/mL) (Sigma-Aldrich, Milan, Italy), added 2 h later than the incubation of the cells with celecoxib or GS, or their combination. 

For the transient transfection procedure, chondrocytes were subjected to a small interfering silencing (siRNA). They were incubated with miR-34a, miR-146a, miR-181a, and miR-210 specific inhibitors (50 nM) (Qiagen, Hilden, Germany), which are antisense oligonucleotides with perfect sequence complementary to their targets; when introduced into cells, with a specific transfection reagent, they sequester the target miRNA in highly stable heteroduplexes, preventing the miRNA from hybridizing with its normal cellular interaction partners. The related negative controls (NC) were used at the concentration of 5 nM (Qiagen, Hilden, Germany). The procedure was conducted for 24 h, according to the kit instructions. Afterwards silencing, the standard treatment with celecoxib or/and GS in presence or not of IL-1β was applied. 

Timing of treatments were chosen on the basis of the best results obtained in terms of viability and according to previous studies [43,86].

Finally, some cultures were processed for 2 h with a specific inhibitor of NF-κB kinase subunit alpha, named IKKα/β (BAY 11-7082, 1 μM) (Sigma–Aldrich, Milan, Italy), according to the manufacturer’s instructions, and then incubated for 24 h with celecoxib or/and GS in presence of IL-1β.

### 4.3. MTT Assay

The viability of the cells was evaluated by MTT assay (3-[4,4-dimethylthiazol-2-yl]-2,5-diphenyl-tetrazoliumbromide) (Sigma–Aldrich, Milan, Italy) and expressed as the percentage of survival. The procedure has been performed according to previously established method [43]. The percentage of survival cells was measured as (absorbance of the sample)/(absorbance of control) × 100 by a microplate reader spectrophotometer at 570 nm (BioTek Instruments, Inc., Winooski, VT, USA). The obtained data were reported as optical density units per 10^4^ adherent cells.

### 4.4. Apoptosis Labeling and Reactive Oxygen Species Assessment

The detection of apoptosis was assessed through a commercial kit provided with annexin-V and propidium iodide (PI) probes (ThermoFisher Scientific, Milan, Italy). After treatment, the cells were harvested, collected into cytometry tubes, and centrifuged. The pellet was resuspended in a working solution of annexin-V and PI, according to the manufacturer’s instructions, and incubated at room temperature for 15 min in the dark. 

The evaluation of mitochondrial superoxide anion and of intracellular ROS production was carried out by using a commercial kit of MitoSOX Red and 2′,7′-Dichlorodihydrofluorescein diacetate (DCFH-DA) probes (ThermoFisher Scientific, Milan, Italy). After treatment, the cells were incubated for 30 min at 37 °C in the dark with a solution of MitoSOX Red, according to the instructions, and then collected into cytometry tubes and centrifuged. The pellet was resuspended in saline solution before the analysis at flow cytometer (Cy Flow Cube 6, Sysmex Partec, Milan, Italy).

A total of 10,000 cells per assay was measured by the instrument both for apoptosis and ROS assessment. The obtained results were analyzed with Cell Quest software (Version 4.0, Becton Dickinson, San Jose, CA, USA). For the detection of apoptotic cells, the analysis was carried out measuring cells simultaneously stained and positive to each dye, and the results were reported as percentage of total apoptosis; for the production of superoxide anion the results were expressed as median fluorescence.

### 4.5. Nitric Oxide Detection

The release of NO in the culture medium of the cells was measured by using the Griess method. One hundred μL of supernatant from each experimental condition was transferred in a microplate with 100 μL of Griess reagent (1% of sulfanilamide, 0.1% of N-1-naphthylethylenediamide dihydrochloride in 5% of H_3_PO_4_) and were incubated at room temperature for 15 min. The absorbance was measured at 550 nm using a microplate reader. The obtained results were normalized with the number of the cells and the NO concentration was reported as ng/10^6^ cells.

### 4.6. Quantitative Real-Time PCR

After treatment, the cells were harvested and total RNA was extracted using TriPure Isolation Reagent (Euroclone, Milan, Italy), according to the manufacturer’s instructions. The concentration, purity, and integrity of RNA were evaluated by Nanodrop-1000 (Celbio, Milan, Italy). 

For the reverse-transcription of target genes and miRNA, 500 ng of RNA were processed into cDNA using specific commercial kits (Qiagen, Germany), and then employed for real-time PCR using specific commercial kits for SYBR Green assay (Qiagen, Germany). The primers used for PCR procedure were reported in Appendix A.

All qPCR reactions were executed in glass capillaries by a Light Cycler 1.0 instrument (Roche Molecular Biochemicals, Milan, Italy) using Light Cycler Software Version 3.5. 

For the data analysis, the Ct values of each sample were calculated and converted into relative quantities. The normalization was performed using housekeeping genes, Actin Beta (ACTB) for target genes and Small Nucleolar RNA, C/D Box 25 (SNORD-25) for miRNA.

### 4.7. Immunofluorescence Determination

For this determination, cells were grown in specific and sterile circles coverslips of borosilicate glass (ThermoFisher Scientific, Milan, Italy) mounted in multi-wells at a starting low density of 4 × 10^4^ cells/chamber, to avoid any possible achievement of confluence or overlapping. Chondrocytes were treated for 24 h with celecoxib and GS in combination, in presence or not of IL-1β. After that, cells were fixed in 4% paraformaldehyde (ThermoFisher Scientific, Milan, Italy) for 15 min, and permeabilized with a blocking solution contained PBS, 1% bovine serum albumin (BSA) (Sigma–Aldrich, Italy) and 0.2% Triton X-100 (ThermoFisher Scientific, Milan, Italy) for 30 min. After these steps, the cells were incubated overnight, at 4 °C, with solutions of mouse monoclonal anti-BCL2 and anti-NRF2 primary antibodies (Santa Cruz Biotechnology, Milan, Italy) (dilution 1:100), followed by 1 h incubation with goat anti-mouse IgG-Texas Red conjugated antibody (Southern Biotechnology, Rome, Italy) (dilution 1:500). After antibodies incubations, the coverslips were washed and a nuclear colorant was added before mounting in specific slides. 

Fluorescence was examined with a Leitz Aristo plan fluorescence microscope and the epifluorescence was analyzed at 200× and 400× magnification. About 100 cells for each experimental conditions were randomly considered and scored by the same operator. The fluorescent signal was evaluated as fair, medium, or strong label [87].

### 4.8. Western Blot Analysis

For protein extraction, cell pellets were suspended in ice-cold lysis buffer [50 mM Tris-HCl (pH 7.5), 50 mM EDTA (pH 8), 150 mM NaCl, 1% NP40, 2 mM Na_3_VO_4_, 10 mM NaF, 0.3 mM PMSF, and a protease inhibitor cocktail (Cat#87785, ThermoFisher Scientific, Milan, Italy)]. Protein concentrations were determined using the Bradford Method (Cat# S-B6916, Sigma-Aldrich, Milan, Italy). 10 μg of proteins were loaded onto a 10% sodium dodecyl sulfate-polyacrylamide gel and then transferred onto a nitrocellulose membrane (Cat# 1620115, Bio-Rad, Hercules, CA, USA). Primary antibodies anti- BCL2 (sc-7382 Mouse Monoclonal antibody, Santa Cruz Biotechnology, Milan, Italy) (dilution 1:100), anti-NRF2 (sc-365949 Mouse Monoclonal antibody, Santa Cruz Biotechnology, Italy) (dilution 1:100) and β-actin (sc-47778 Mouse Monoclonal antibody, Santa Cruz Biotechnology, Milan, Italy) (dilution 1:10,000) were incubated overnight at 4 °C. Membranes were washed with TBS with 0.1% Tween-20 and incubated with horseradish peroxidase-conjugated secondary antibodies (dilution 1:1000) for 1 h at room temperature. Then, membranes were washed before chemiluminescence detection using Clarity ECL reagents (Cat #1705061, Bio-Rad, Hercules, CA, USA) and the images were acquired with Chemidoc MP (Bio-Rad, Hercules, CA, USA). Image Lab software version 6.1 (Bio-Rad, Hercules, CA, USA) was used to quantify the images of the bands. Results were normalized with the relative loading control.

### 4.9. Statistical Analysis

Three independent experiments were carried out and the results were expressed as the mean ± standard deviation of triplicate values for each experiment. Data normal distribution was evaluated by Shapiro–Wilk, D’Agostino and Pearson, and Kolmogorov–Smirnov tests. Statistical analysis was performed using analysis of variance followed by the Bonferroni multiple comparison test. All analyses were carried out through the SAS System (SAS Institute Inc., Cary, NC, USA) and GraphPad Prism version 6.01/b for Windows (GraphPad Software version 6.01/b, Boston, MA, USA, www.graphpad.com, accessed on 10 february 2021). A *p*-value < 0.05 will be defined as statistically significant.

## 5. Conclusions

In the present study, we confirmed, first, the regulatory synergistic effects of celecoxib and prescription-grade GS on apoptosis process and oxidant system, in human OA chondrocyte cultures. 

In addition, we proved, for the first time, the ability of the studied compounds to modulate the expression profile of some miRNA, known to be implicated in apoptosis and oxidative stress processes, and, therefore, in OA pathophysiology. 

Finally, we identified miR-34a, miR-146a, and miR-210 as possible mediators of celecoxib and GS-induced positive effects, via NF-κB pathway, providing additional information about their molecular mechanism. Furthermore, these data support the potential role of miRNA as therapeutic targets for the treatment of OA [88].

This study confirms the synergistic anti-inflammatory and chondroprotective effects of celecoxib and GS, as previously demonstrated in our in vitro experience, and support the therapeutic use of this combination in the multimodal approach for patient with OA [89].

Anyway, further studies are required to better comprehend this complex network, such as the identification of the direct targets of miRNA, controlling apoptosis and oxidative stress signaling, which may help to elucidate the downstream cascade triggered by celecoxib and GS. Furthermore, considering that the studied miRNA also resulted to be implicated in regulating cartilage turnover, an additional analysis evaluating the effect of miRNA silencing on the expressional levels of the matrix degrading enzymes, such as metalloproteinases, or matrix proteins, as collagen and proteoglycans, should be performed; therefore, the potential influence of celecoxib and GS on cartilage homeostasis after miRNA silencing could be an important aspect to investigate in the future.

## Figures and Tables

**Figure 1 ijms-24-14994-f001:**
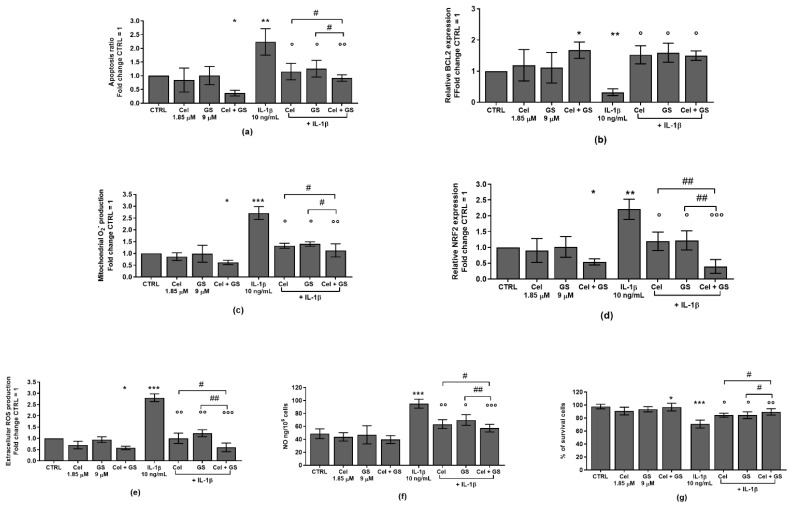
Chondrocytes were evaluated at control (CTRL) condition, after 24 h of treatment with celecoxib (cel) (1.85 µM) and glucosamine sulfate (GS) (9 µM), in presence or not of interleukin (IL)-1β (10 ng/mL). (**a**,**c**,**e**) Apoptosis detection and reactive oxygen species (ROS) production by flow cytometry. (**b**,**d**) Expression levels of B-cell lymphoma 2 (*BCL2*) and nuclear factor erythroid 2 (*NRF2*) by quantitative real time PCR. (**f**) Nitric oxide (NO) release by Griess method. (**g**) Viability by MTT assay. All the results, except for NO and viability, were expressed as fold change of the value of interest respect to CTRL, reported equal to 1. Data were represented as mean ± standard deviation. * *p* < 0.05, ** *p* < 0.01, *** *p* < 0.001 versus CTRL. ° *p* < 0.05, °° *p* < 0.01, °°° *p* < 0.001 versus IL-1β. # *p* < 0.05, ## *p* < 0.01 versus Cel or GS + IL-1β.

**Figure 2 ijms-24-14994-f002:**
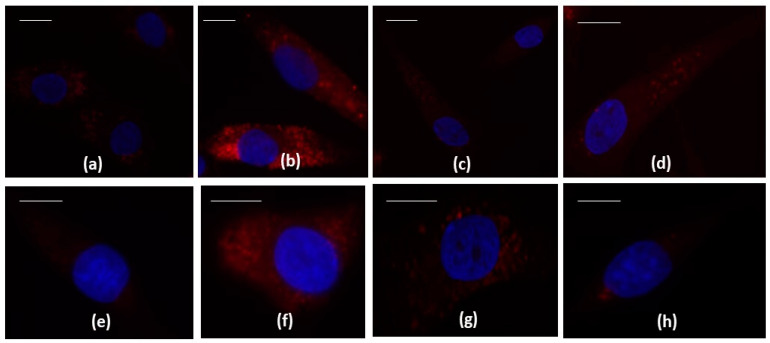
Chondrocytes were evaluated at control (CTRL) condition, after 24 h of treatment with celecoxib (cel) (1.85 µM) and glucosamine sulfate (GS) (9 µM), in presence or not of interleukin (IL)-1β (10 ng/mL). (**a**–**h**) Indirect immunofluorescence microscopy of cells incubated with monoclonal anti- B-cell lymphoma 2 (BCL2) (**a**–**d**) and anti-nuclear factor erythroid 2 (NRF2) (**e**–**h**) primary antibodies. (**a**,**e**) CTRL: a fair fluorescence in the cytoplasm is shown; (**b**,**f**) IL-1β: an intense signal is evident in the cytoplasm; (**c**,**g**) Cel + GS; the label is almost absent. (**d**,**h**) Cel + GS + IL-1β: the signal is diffused in the cytoplasm but reduced with respect to IL-1β. Nuclei (blue) were stained with DAPI. Bars: (**a**–**h**) 50 µm.

**Figure 3 ijms-24-14994-f003:**
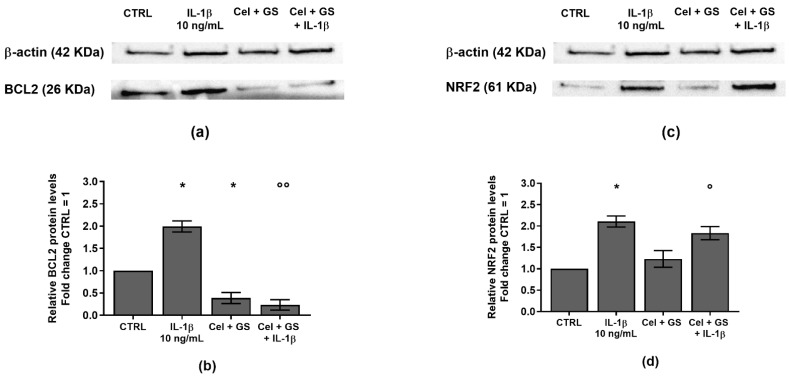
Chondrocytes were evaluated at control (CTRL) condition, after 24 h of treatment with celecoxib (cel) (1.85 µM) and glucosamine sulfate (GS) (9 µM), in presence or not of interleukin (IL)-1β (10 ng/mL). (**a**–**d**) Representative immunoblotting image and densitometric analysis of B-cell lymphoma 2 (BCL2) and nuclear factor erythroid 2 (NRF2) protein levels by western blot. The results were expressed as fold change of the value of interest respect to CTRL, reported equal to 1. Data were represented as mean ± standard deviation. * *p* < 0.05 versus basal CTRL. ° *p* < 0.05, °° *p* < 0.01 versus IL-1β.

**Figure 4 ijms-24-14994-f004:**
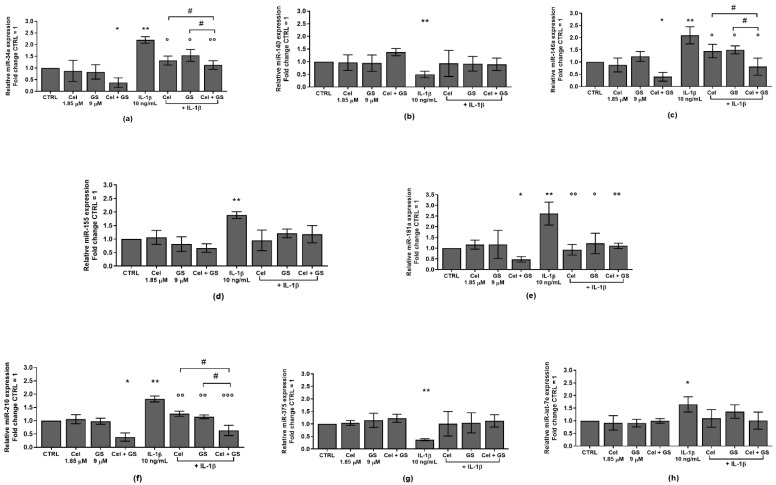
Chondrocytes were evaluated at control (CTRL) condition, after 24 h of treatment with celecoxib (cel) (1.85 µM) and glucosamine sulfate (GS) (9 µM), in presence or not of interleukin (IL)-1β (10 ng/mL). (**a**–**h**) Expression levels of microRNA by quantitative real time PCR. The results were expressed as fold change of the value of interest respect to CTRL, reported equal to 1. Data were represented as mean ± standard deviation. * *p* < 0.05, ** *p* < 0.01 versus basal CTRL. ° *p* < 0.05, °° *p* < 0.01, °°° *p* < 0.001 versus IL-1β. # *p* < 0.05 versus Cel or GS + IL-1β.

**Figure 5 ijms-24-14994-f005:**
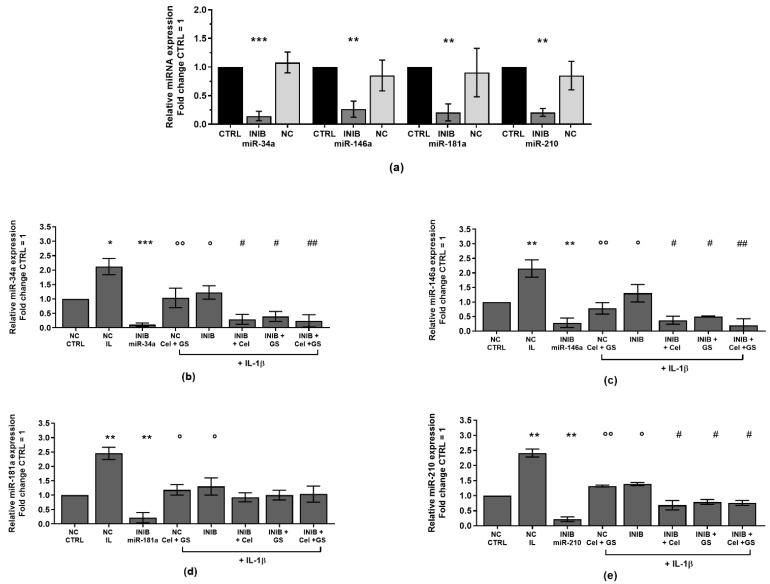
Chondrocytes were evaluated at control (CTRL) condition, after 24 h of transient transfection with miR-34a, miR-146a, miR-181a, and miR-210 inhibitors (50 nM) or NC (5 nM), 24 h of incubation with celecoxib (cel) (1.85 µM) and glucosamine sulfate (GS) (9 µM), in presence of interleukin (IL)-1β (10 ng/mL). (**a**–**e**) Expression levels of microRNA by quantitative real time PCR. The results were expressed as fold change of the value of interest respect to CTRL, reported equal to 1. Data were represented as mean ± standard deviation. * *p* < 0.05, ** *p* < 0.01, *** *p* < 0.001 versus CTRL or NC. ° *p* < 0.05, °° *p* < 0.01 versus IL-1β. # *p* < 0.05, ## *p* < 0.01 versus cel + IL, GS + IL and cel + GS + IL. INIB = Inhibitor, NC = negative control siRNA.

**Figure 6 ijms-24-14994-f006:**
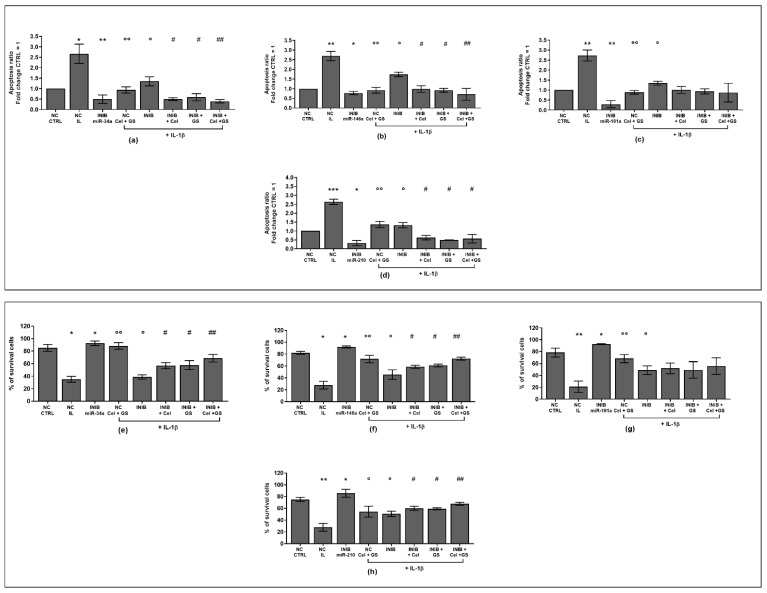
Chondrocytes were evaluated at control (CTRL) condition, after 24 h of transient transfection with miR-34a, miR-146a, miR-181a, and miR-210 inhibitors (50 nM) or NC (5 nM), 24 h of incubation with celecoxib (cel) (1.85 µM) and glucosamine sulfate (GS) (9 µM), in presence of interleukin (IL)-1β (10 ng/mL). (**a**–**d**) Apoptosis detection by flow cytometry. The results were expressed as fold change of the value of interest respect to CTRL, reported equal to 1. (**e**–**h**) Viability by MTT assay. Data were represented as mean ± standard deviation. * *p* < 0.05, ** *p* < 0.01, *** *p* < 0.001 versus CTRL or NC. ° *p* < 0.05, °° *p* < 0.01 versus IL-1β. # *p* < 0.05, ## *p* < 0.01 versus cel + IL, GS + IL and cel + GS + IL. INIB = Inhibitor, NC = negative control siRNA.

**Figure 7 ijms-24-14994-f007:**
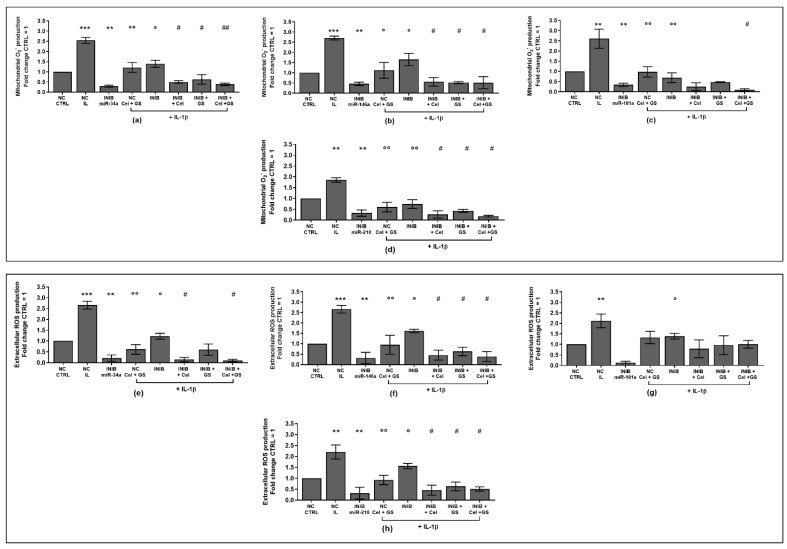
Chondrocytes were evaluated at control (CTRL) condition, after 24 h of transient transfection with miR-34a, miR-146a, miR-181a, and miR-210 inhibitors (50 nM) or NC (5 nM), 24 h of incubation with celecoxib (cel) (1.85 µM) and glucosamine sulfate (GS) (9 µM), in presence of interleukin (IL)-1β (10 ng/mL). (**a**–**d**) Mitochondrial superoxide anion and (**e**–**h**) extracellular ROS production by flow cytometry. The results were expressed as fold change of the value of interest respect to CTRL, reported equal to 1. Data were represented as mean ± standard deviation. ** *p* < 0.01, *** *p* < 0.001 versus CTRL or NC. ° *p* < 0.05, °° *p* < 0.01 versus IL-1β. # *p* < 0.05, ## *p* < 0.01 versus cel + IL, GS + IL and cel + GS + IL. INIB = Inhibitor, NC = negative control siRNA.

**Figure 8 ijms-24-14994-f008:**
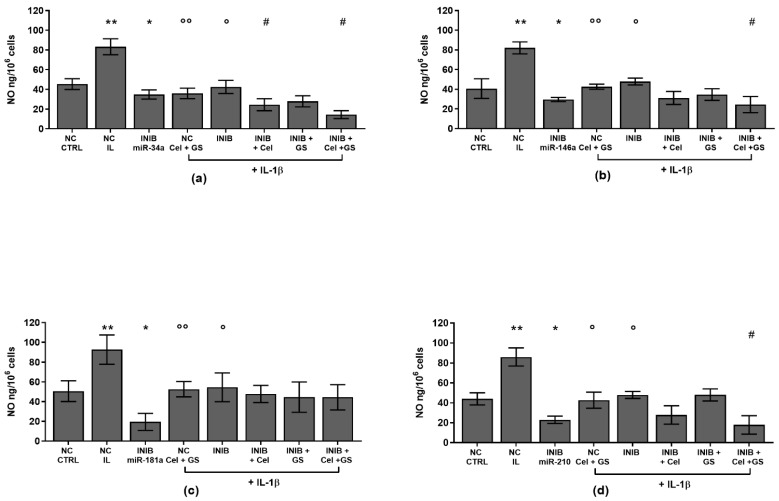
Chondrocytes were evaluated at control (CTRL) condition, after 24 h of transient transfection with miR-34a, miR-146a, miR-181a, and miR-210 inhibitors (50 nM) or NC (5 nM), 24 h of incubation with celecoxib (cel) (1.85 µM) and glucosamine sulfate (GS) (9 µM), in presence of interleukin (IL)-1β (10 ng/mL). (**a**–**d**) Nitric oxide (NO) release by Griess method. Data were represented as mean ± standard deviation. * *p* < 0.05, ** *p* < 0.01 versus CTRL or NC. ° *p* < 0.05, °° *p* < 0.01 versus IL-1β. # *p* < 0.05 versus cel + IL, GS + IL and cel + GS + IL. INIB = Inhibitor, NC = negative control siRNA.

**Figure 9 ijms-24-14994-f009:**
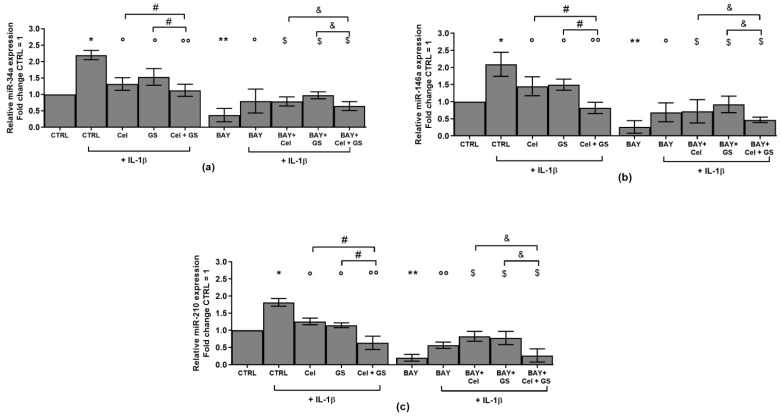
Chondrocytes were evaluated at control (CTRL) condition, after 24 h of incubation with a specific nuclear factor (NF)-κB inhibitor (BAY 11-7082, IKKα/β, 1 μM), 24 h of treatment with celecoxib (cel) (1.85 µM) and glucosamine sulfate (GS) (9 µM), in presence of interleukin (IL)-1β (10 ng/mL). (**a**–**c**) Expression levels of microRNA by quantitative real time PCR. The results were expressed as fold change of the value of interest respect to CTRL, reported equal to 1. Data were represented as mean ± standard deviation. * *p* < 0.05, ** *p* < 0.01 versus CTRL. ° *p* < 0.05, °° *p* < 0.01 versus IL-1β. # *p* < 0.05 versus cel or GS + IL-1β. ^$^
*p* < 0.05 versus BAY + IL-1β; ^&^
*p* < 0.05 versus BAY + Cel or GS + IL-1β.

## Data Availability

The data used and/or analyzed during the current study are available from the corresponding author on reasonable request.

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
