# Peer review of "MicroRNA as Possible Mediators of the Synergistic Effect of Celecoxib and Glucosamine Sulfate in Human Osteoarthritic Chondrocyte Exposed to IL-1β"

_ijms, 2023, doi:10.3390/ijms241914994_

Round 1

Reviewer 1 Report

This is a thought-provoking article, and I would like to offer some suggestions for improvement: 

1.     Please remove the unnecessary spaces in lines 22, 28, and 33.

2.     In line 24, replace "MTT" with the full term, and similarly, in line 27, write out "PCR." In lines 33 and 37, provide the full form for "NF-kB," and for "microRNA" in line 37.

3.     Regarding the role of miRNA in the pathogenesis of OA, consider updating reference 2 with the following recent references:

https://www.mdpi.com/1422-0067/24/17/13144

https://www.mdpi.com/2227-9059/11/4/1189

4.     When NRF2 first appears in line 100, briefly explain its significance, similar to the explanation given for BCL2 in line 98.

5.     Kindly furnish a concise elucidation detailing the methodology employed for quantifying the proportion of cells expressing cytoplasmic BCL2 (refer to lines 121-126) and NRF2 (refer to lines 127-133) in section 4.7. If image analysis software was utilized in this process, kindly specify its nomenclature. Additionally, elucidate the procedure followed in selecting 100 cells as outlined in line 518, ensuring that this selection was genuinely randomized.

6.     Revise the expression of p-values throughout the "2. Results" section for clarity and consistency.

7.     Clarify the rationale behind selecting the specific set of miRNAs in the pathogenesis of OA, either in the introduction or section 2.2.

8.     Rewrite lines 310-312, as the second instance of "BCL2" appears to be a potential mistake.

The English is passable but not in supreme quality. 

Author Response

Please see the aattachment

Reviewer 2 Report

The manuscript is interesting, even if the first part of the data has been published in a previous paper https://doi.org/10.3390/ijms22168980

My comments are as follows:

Introduction

The focus of the manuscript is to study the effect of the celecoxib and GS, alone or in combination, on the expression profile of a group of miRNAs in OA chondrocytes but there is no background/information on the pathology in the introduction. A brief introduction on OA should be added. It should be specified that OA is a whole joint disease involving all joint tissues (infrapatellar fat pad, synovial membrane, meniscus, subchondral bone and cartilage).

Considering also that the journal is not specific on rheumatological/orthopedic diseases, an introduction on cartilage and OA should be added. It should be explained that OA cartilage appears disrupted with an altered composition determining changes in the mechanical and biochemical function etc.

Lines 43-45: references should be provided.

Line 56, 229: contracted forms like “isn’t” or “didn’t” should be not used in a scientific manuscript.

Line 61: “ESCEO” should be defined.

Lines 84-85: could the authors better explain why they select to evaluate these miRNAs?

Results

Letters of the figures should be lowercase at the bottom center in brackets.

Figure 1: looking at the figure, it seems that there is no difference between IL1 beta + Cel alone vs IL1beta+ GS alone and IL1beta + Cel + GS. Is that correct? If there is no difference, it cannot be concluded that there is a synergistic effect of cel + GS.

Figure 2 needs to be improved. It is difficult to see cells. Moreover, the labels should be improved in order to understand the different conditions and the antibody used.

Lines 192: what kind of inhibitors did the authors use?

It is unclear why figure s1 is reported in the manuscript and not in the supplementary file.

Figure s1E: the inhibition of all miRNAs determines a significant decrease of NO production with the only exception of miR-210. The authors should check.

Figure s1D: Is there a significant difference between NC and CTRL of miR-181a?

Lines 216-220: the authors reported that silencing of chondrocytes with miR-34a, miR-146a, miR-181a, and miR-210 inhibitors induced an increase of viability (p < 0.05). this is correct for all excepted miR-210. The authors should check figure s1.  

Figure s1 and figure 6: viability of IN miR-210 is not significant (compared to control) In figure 5, while it is significant in figure 6 it is significant.

Figure 9: could the authors add cel alone and GS alone treatments?

It would be useful to see also the effects of miRNA inhibition and treatments on MMPs, collagens etc.

Discussion

Looking at the data it seems that the combination of both drugs (GS + Cel) does not give better results than the individual drugs. This point needs to be discussed.

Conclusion

The conclusions should reflect the results. It is questionable to conclude that there is a synergistic effect of celecoxib and GS.  

Methods

Line 443: Could the authors better specify miR-34a , miR-146a, miR-181a, and miR-210 specific inhibitors? The authors reported that they used siRNA (NC) (5 nM) as negative control. Thus, I suppose that the authors used siRNA to block miRNAs? The authors should clarify this part.

Line 454: supplier of MTT assay should be added. Microplate reader used should be specified.

Line 471: Flow cytometer used should be added. Software used for the analysis should be specified.

Line 505: “specific and sterile coverslips” should be specified.

Line 513, 527-531: dilution of the antibodies should be added.

Line 544: graphpad should be cited as follows: https://www.graphpad.com/guides/prism/latest/user-guide/citing_graphpad_prism.htm

Other comments

It is not clear how authors did the experiments without any funds. How did the authors purchase the reagents without funds?

Round 2

Reviewer 1 Report

The revised work demonstrates notable improvement. Allow me to point out some minor details:

1. The unnecessary space in line 34 should be removed.

2. The use of the "$" symbol in Figure 9 might be slightly misleading. It would be advisable to employ distinct symbols for the "$" with and without brackets within the figures. 

3. In line 118, kindly reference the following current articles introducing NRF2:

https://www.mdpi.com/1422-0067/24/13/10872

https://www.mdpi.com/2227-9059/11/6/1512

https://www.mdpi.com/2075-1729/13/4/1045

4. Line 198: “mir-181a” ought to be corrected to “miR-181a”.

Reviewer 2 Report

I have still some comments for the authors.

Lines 45-48: The authors should add also the infrapatellar fat pad. This tissue is inflamed and fibrotic in OA patients (DOI:10.1093/rheumatology/kex287).

Lines 49-52: the biomechanical behaviour of cartilage and chondrocytes changes in OA (DOI: 10.3390/pr11041014 etc)

Lines 53-56: the authors forgot to mention that also infrapatellar fat pad secretes pro-inflammatory molecules contributing to exacerbate OA (https://doi.org/10.1002/jor.25347 etc)

Previously, I asked to properly cite Graphpad as explained in the website of the software: https://www.graphpad.com/guides/prism/latest/user-guide/citing_graphpad_prism.htm. I mean that the authors need to check the website and not cite it.

Funding provided by the department should be declared.
